# Influence of Heat Treatment on the Corrosion Resistance of Aluminum-Copper Coating

**Mieczyslaw Scendo [1],* , Slawomir Spadlo [2], Katarzyna Staszewska-Samson [1] and Piotr Mlynarczyk [2]**

[1] Institute of Chemistry, Jan Kochanowski University in Kielce, Uniwersytecka 15G, PL-25406 Kielce, Poland; katarzyna.staszewska@onet.pl

[2] Department of Computer Techniques and Armaments, Kielce University of Technology, Tysiaclecia Panstwa Polskiego 7, PL-25314 Kielce, Poland; sspadl@tu.kielce.pl or sspadlo@onet.pl (S.S.); piotrm@tu.kielce.pl (P.M.)

* Correspondence: scendo@ujk.edu.pl; Tel.: +41-349-7045; Fax: +41-349-7062

**Abstract:** Influence of heat treatment on the corrosion resistance of the aluminum-copper (Al-Cu) coating on the aluminum substrate was investigated. The coating was produced by the electrical discharge alloying (EDA) method. The surface and microstructure of the specimens were observed by a scanning electron microscope (SEM). The phase analysis of the composite materials by X-ray diffraction (XRD) and energy-dispersive spectroscopy (EDS) indicated that intermetallic compounds (i.e., $CuAl_2$ and $Cu_9Al_4$) were formed through reactions between Al and Cu. during the EDA process. A significant increase in the hardness of the Al-Cu coating was affected by the improvement of the alloy structure. The heat treatment of materials was carried out at 400 °C or 600 °C in the air atmosphere. A corrosion test of materials was carried out by using electrochemical methods. The corrosive environment was acidic chloride solution. After heat treatment at 400 °C the mechanical properties of the Al/Cu alloy increased significantly and the oxide layer protect of the alloy surface against corrosion. However, after heat treatment at elevated temperature, i.e., 600 °C it was found that the $(Al_2O_3)_{ads}$ and $(CuO)_{ads}$ coatings were destroyed. The mechanical properties of the Al/Cu alloy decreased, and its surface has undergone deep electrochemical corrosion.

**Keywords:** electrical discharge alloying; copper coating; heat treatment; acidic chloride solution; corrosion parameters; corrosion rate

## 1. Introduction

Electrical discharge alloying (EDA) is a new surface alloying method that uses a composite electrode as an alloying material to improve the surface chemical and mechanical properties of structural materials. The EDA method is carried out using short-lived, high-current electrical pulses with a very low total heat input [1,2]. The EDA method achieves a metallurgical bonded coating on a metal substrate. The short duration of the electric impulse allows for extremely fast solidification of the deposited material and the formation of a fine-grained, homogeneous coating with an almost amorphous structure. The obtained EDA method coating has exceptionally good tribological and anticorrosion properties [3,4]. However, during the EDA process the release of capacitor energy will generate a high temperature plasma arc between the electrode tip and the substrate. The electrode material is ionized by the plasma arc and the molten electrode material is transferred onto the substrate. The scheme of the electrical discharge alloying coating deposition on a metal substrate is shown in Figure 1. Moreover, additional information on the mathematical model of the EDA process can be found in article [5].

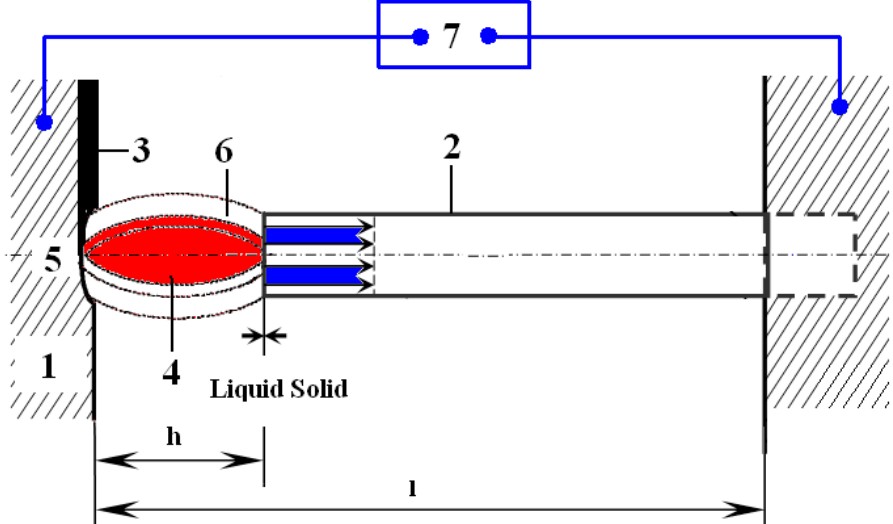

**Figure 1.** Scheme the electrical discharge alloying coating deposition: 1—substrate (cathode), 2—erode as coating material (anode), 3—coating, 4—plasma, 5—diffusion/reaction zone, 6—protective gas, 7—power generator, h—height plasma and l—length erode.

In the initial stage of deposition, physical and chemical processes take place in the inter-electrode gap (called the frontal) and the substrate. The EDA technology relies on the phenomenon of electrical discharge, which is accompanied by mass transport and energy dissipation. As a result of erosion of the working electrode (erode) the electrode particles deposit on a metal substrate (cathode). Moreover, the deposition of the coating takes place with the simultaneous heating of the substrate at the contact point of both electrodes. In this area, local melting of the electrodes occurs, so-called electrical discharge. Furthermore, the electrical discharge process is associated with the ionization and heating of the air to a high temperature of up to ten thousand degrees. However, the phase composition and properties of the deposited coatings depend on the type of substrate material and the electrode. The generated composite coating shows improved hardness and protects the substrate against wear. It is the result of interactions between the material of the erode and the metallic substrate. It is worth mentioning that in some cases discharge coatings are more roughened than the substrate. Moreover, the tensile stresses may be created on the coated surface, which significantly reduce the fatigue strength of the materials. Undesirable effects of the EDA coating can be minimized or eliminated by mechanical or laser treatment [3–5].

Low-density aluminum has a relatively lower hardness than other metals. Therefore, metals such as: Fe, Mg, Mo or Ti were used to improve the mechanical properties of aluminum [6–9]. Moreover, various methods have been used to obtain a good quality coating of various metals on the Al surface. In addition, good adhesion of the tested metals to the aluminum surface was obtained. To increase the hardness of Al, its surface was covered with more hard composites, e.g., carbon nanotubes (CNT) [10]. It was found that the increase in the mechanical characteristics of Al was attributed to the effects of the particular strengthening by CNT and the regularly oriented carbon nanotubes achieved through the nanoscale dispersion method. Recently, interest in aluminum and copper composites was increasing, as these composites combine the lightweight properties of Al and the thermal characteristics of Cu. The accumulative roll bonding (ARB) is one of the latest methods used to produce of the Al/Cu composite material [11]. It turned out that the ARB method can achieve a copper coating on a metal substrate in this case the average thickness of Cu layer decreased from 100 to 7 µm. It is worth noting that the strength and hardness of the copper coating clearly increased. However, the authors of the work [12] in order to obtain the Al-Cu/DLC (diamond like carbon) composite applied the squeeze casting (SC) method. It was reported that the thermal conductivity of the investigated composite increased significantly. Moreover, the aluminum/copper composite

materials were successfully fabricated by the spark plasma sintering (SPS) method [13]. It was found that of the Al/Cu mechanical properties have improved considerably as well as the copper content affects physical properties of the Al/Cu composite material. Moreover, an increase in the amount of intermetallic compounds was found in the composite material.

Recently, research on the deformation behavior and precipitation features of Al/Cu alloys were carried out. Among others, the authors of the work [14] researched of high temperature compressive features of the Al/Cu alloy. They showed that a high temperature significantly reduces the effect of compressive features of the alloy. Liu et al. [15] described the continuous dynamic recrystallization behavior of a compressed Al/Cu alloy at the temperature range of 350–500 °C. Whereas the authors of the work [16] discussed the effect of a cooling rate and solution time on the precipitated transformation of Al/Cu alloy, and they found that the precipitation of the new phases was enhanced with increased cooling rate and solution time. Moreover, deformation behavior and precipitation features of the Al/Cu alloy were investigated using uniaxial tensile tests at intermediate temperatures [17]. It was found that the true stress drops with the decreased strain rate or the increased deformation temperature.

The properties of the copper/aluminum (Cu/Al) clad and copper/aluminum/copper (Cu/Al/Cu) laminated composites have been studied by [18–20]. Both composites were fabricated by the cold rolling method (CR) [21,22]. Their advantages associated with high conductivity, low density and good surface performance. Therefore, the Cu/Al and Cu/Al/Cu composites have been successfully utilized in the fields of electrical and electronic components [23,24].

Accordingly, many efforts have recently been made to develop surface modification or the production of new surface materials whose properties differ from those of the substrate. The most efficient and recommended for the production of metallic protective coatings is the electrical discharge alloying method, because: (i) it is possible to produce thin (i.e., number μm) or thick (i.e., number mm) metallic coatings, (ii) metallic coatings are characterized by good adhesion with the substrate and (iii) metallic coatings can be applied in strictly defined places and on complicated shapes. Moreover, in some cases the EDA coatings are rough. In addition, tensile stresses can be created on the surface of the coating, which significantly reduce the fatigue strength of the materials.

Aluminum is very useful in fabricating lightweight structures. However, the tribological applications of these materials are limited. This paper presents a brief study on improving the mechanical properties and corrosion resistance of aluminum by covering the surface with a copper layer using the electrical discharge alloying method. However, there is no information in the literature regarding the anticorrosive properties of the Al-Cu coatings that were produced by the EDA method.

In the present study, the influence of the heat treatment (400 °C or 600 °C) on the corrosion resistance of the Al-Cu coating was investigated. The coating was produced on the Al substrate by the electrical discharge alloying (EDA) method. Before and after heat treatment the microstructure, mechanical properties and corrosion resistance of the materials were carried out by using of the electrochemical methods.

## 2. Materials and Methods

The aluminum-copper coatings on the aluminum (99.8% Al) substrate were produced by the electrical discharge alloying method. For this purpose, a homemade energy generator was used. During the EDA treatment 600 V electrical impulses were used, while the capacitors battery had a capacity of 250 μF. The discharge current was 5 A. The electric discharge time was 20 milliseconds. The source of the coating material was a specially crafted copper (99.8% Cu) electrode (erode) and the diameter of the copper wire was 5 mm. Unfortunately, the chemical composition of the copper electrode lagging is unknown. What is more, helium (99.8% He) was used as a protective gas. However, the coating production efficiency was 150–170 mg cm$^{-2}$.

Figure 2 shows a photograph image of the Al/Cu alloy surface produced by the electrical discharge alloying (EDA) method. It is clear that the surface of the Al/Cu alloy was rough, but there were no

microcracks, scratches and delamination. Moreover, all copper layers were applied three times on the aluminum surface. The standard thickness of the Al-Cu coatings ranged from 30 to 50 μm.

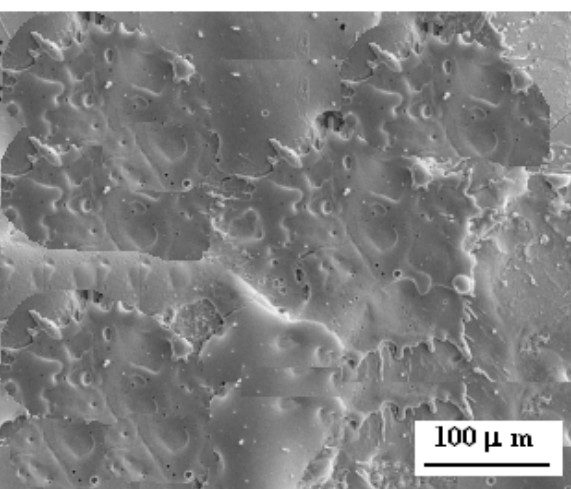

**Figure 2.** Photograph image of Al/Cu surface produced by the electrical discharge alloying method.

The cross-section and microstructure of the specimens were observed by using a scanning electron microscope (SEM) Joel (Tokyo, Japan), type JSM-5400. The accelerating voltage of SEM was 20 kV. The chemical composition for the alloy surface was also measured by an energy-dispersive spectrometer (EDS).

The X-ray patterns diffraction (XRD; Carl Zeiss Jena, Germany), type HZG4 was used to characterize the phase composition of materials the monochromatic Cu, *Kα* radiation source was 0.15407 nm at 30 kV and 15 mA with a step scan mode at intervals of 0.05° in 2θ.

The measurement of microhardness of the tested materials was measured by the Vickers method, using the Falcon 500 hardness tester the INNOVATEST company (Maastricht, The Netherlands). An indenter was used in the form of a diamond pyramid with a square base and an angle between opposite walls equal to 136° whose was load varied from 0.02 to 20 N. The depth of indentation was about 2 μm.

For the heat treatment (thermogravimetric test) of the Al/Cu alloy an electric chamber furnace (CZYLOK, Jastrzebie Zdroj, Poland), type FCF 2.5 HM was used. The heat treatment of materials was carried out in air atmosphere at 400 °C or 600 °C. However, the heat treatment temperature of the specimens was below the melting point of aluminum, i.e., 660 °C. The weight change of the samples was monitored each time after one hour of heating. The heat treatment duration of the Al/Cu alloy was 9 h.

All electrochemical measurements (corrosion tests) were carried out by using PGSTAT 128N (AutoLab, Amsterdam, The Netherlands) potentiostat/galvanostat, piloted by NOVA 1.7 software (AutoLab, Amsterdam, The Netherlands). The electrochemical experiments were carried out in a conventional three-electrode cell.

The working electrode (stationary) was made of the Al/Cu alloy without and after heat treatment at 400 °C or 600 °C. The electrode material was mounted in a special holder. The geometric surface area of the working electrode was 1 cm$^2$. Before each measurement the surface of the electrode was washed with bidistilled water, ultrasonically and dried at room temperature. Subsequently, the working electrode was immediately immersed in the test solution until a steady state was reached. The experiment was started after 30 min of immersion of the electrode in the corrosive solution.

The saturated calomel electrode (SCE) was used as the reference and the counter electrode (9 cm$^2$) was made from platinum foil (99.9% Pt).

The corrosive environment (supporting electrolyte) was obtained by mixing the sodium chloride (NaCl; POCH, Poland) and hydrochloric acid (HCl; POCH, Poland). The concentration of the $Cl^-$ ion was 1.2 M. The solvent used was three distilled water. The pH value was 1.5. The electrolyte was not deoxygenated.

The potentiodynamic polarization (*LSV*) curves were recorded. All measurements were carried out under a potential range from −1200 to −200 mV vs. SCE whereas the potential change rate was 1 mV s$^{-1}$ with holding time of 30 s at −1200 mV. In this way, the surface of the working electrode was cleaned because the metal oxides adsorbed on the surface were reduced.

The LSV curves were used to designate the corrosion electrochemical parameters, i.e., corrosion potential ($E_{corr}$), corrosion current density ($j_{corr}$) and slope a cathodic ($b_c$) and anodic ($b_a$) branches of polarization curves. However, to determine the corrosion parameters and polarization resistance of the tested materials the Tafel Slope Analysis was used. More information about the Tafel method can be found in our publications [25,26]. However, for large cathodic overpotentials ($\eta/b_c << -1$) the Tafel equation for the cathodic reaction is given by:

$$\eta = log \ (j_{corr}) - b_c \ log \ |j| \tag{1}$$

Analogously, for large anodic overpotentials ($\eta/b_a >> 1$) the Tafel equation for the anodic reaction is:

$$\eta = log \ (j_{corr}) + b_a \ log \ (j) \tag{2}$$

where $\eta$ is defined as the difference between the applied potential and the corrosion potential (i.e., $\eta = E - E_{corr}$) and $j$ is the measured current density. The Tafel equations predict a straight line for the variation of the logarithm of current density with potential (Tafel plots).

The electrochemical corrosion rate of materials were appointed using the following equation:

$$v_{corr} = 3.268 \times \frac{j_{corr} \ M}{n \ \rho} \tag{3}$$

where $j_{corr}$ is the corrosion current density (which was determined by the Tafel slope analysis method), $M$ is the molecular weight of reacting substrate, $n$ is the number of electrons exchanged and $\rho$ is the density of material.

The corrosion electrochemical parameters for the tested materials were determined as the average value based on three measurements.

The chronoamperometric curves (*ChA*) were obtained for the potential values, which were selected on the basis of the potentiodynamic polarization curves. The values of the working electrode potential were carefully selected so as to observe a change in the current density values for the characteristic points on the LSV curves. For this purpose, three values of the working electrode potential were chosen for each material tested (i.e., one potential value concerned the cathodic process and two the anodic process). On this basis, it is possible to define the anticorrosive effect of metal coatings, in this case of the Al-Cu coatings on the aluminum substrate.

All electrochemical measurements were carried out at a temperature of 25 ± 0.5 °C, which were maintained using an air thermostat.

## 3. Results and Discussion

### 3.1. Microstructure of Materials

Figure 3 shows an example the metallographic microstructure of cross-section of the Al/Cu alloy and the results of X-ray microanalysis (with the line scanning) of the chemical elements distribution of the alloy and substrate.

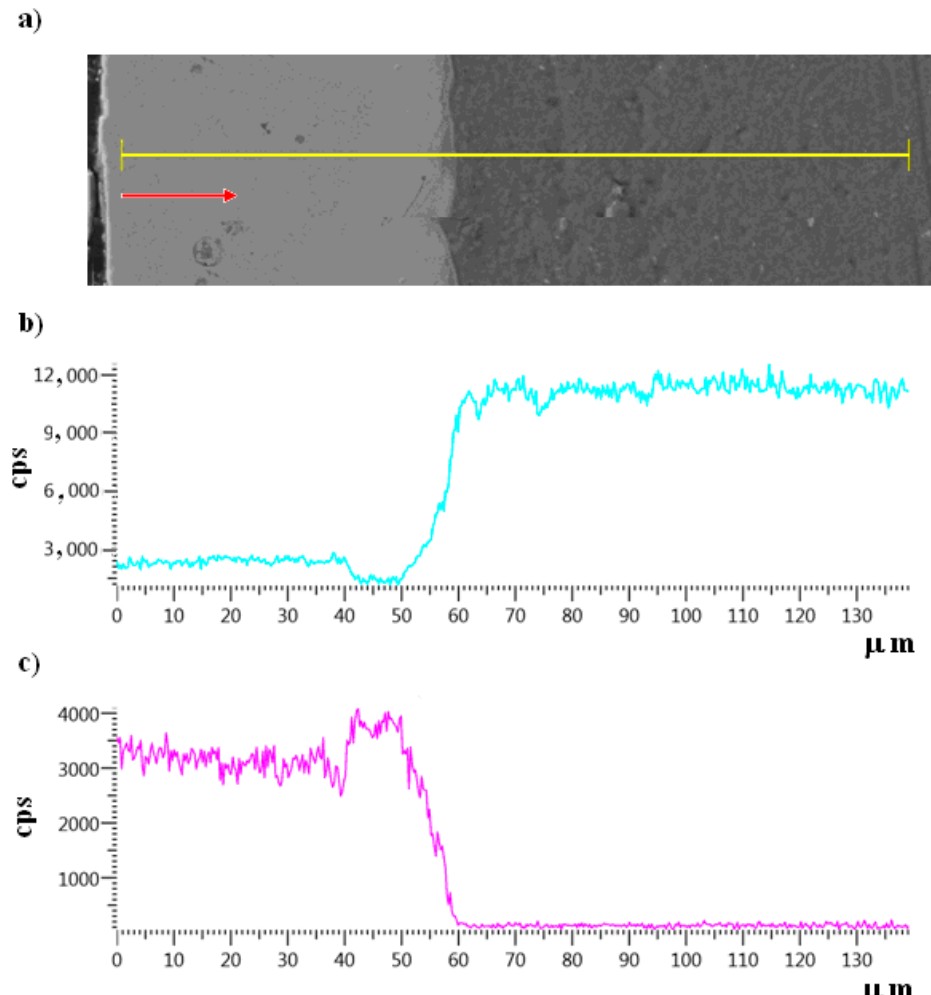

**Figure 3.** X-ray microanalysis of cross-section of Al/Cu alloy: (**a**) metallographic microstructure (with the line scanning) and content distribution: (**b**) aluminum and (**c**) copper.

It turned out that by using the EDA method, diffused copper coatings on the aluminum surface were obtained. What is more, the Al/Cu layer adhered well to the Al substrate. Figure 4 depicts the SEM/EDS image of cross-section of the Al/Cu alloy and the results of point X-ray microanalysis of the chemical composition of the tested alloy.

It was found that of the Al/Cu alloy contained an average of 31% Al and 68% Cu (Figure 4). It is worth adding that the Al-Cu coating contained trace amounts of other elements, i.e., Zn, Cd and Mg. It should be assumed that a large part of copper in the tested layer would significantly improve the corrosion resistance and mechanical properties of aluminum.

Figure 5 presents the XRD patterns for the Al/Cu alloy. Intermetallic compounds composed of the Al phase and Cu phase as well as $CuAl_2$ and $Cu_9Al_4$ phases were detected in the XRD patterns of the Al/Cu alloy. Moreover, in the Al/Cu alloy the natural air-formed oxide layer on the aluminum surface was removed by the micro-plasma generated between particles during the electrical discharge alloying process. In addition, the formation of intermetallic compounds was induced by the activation of intermetallic reactions by local high temperatures.

Therefore, in high temperature conditions (between 400 and 1200 °C) the following chemical reactions are possible on the aluminum surface:

$$2\,Al + Cu \rightarrow CuAl_2 \tag{4}$$

$$Al + Cu \rightarrow CuAl \tag{5}$$

$$4\,Al + 9\,Cu \rightarrow Cu_9Al_4 \tag{6}$$

| Spectrum | Al | Cu |
|---|---|---|
| label | Weight % | |
| Spectrum 1 | 34.0 | 65.2 |
| Spectrum 2 | 35.2 | 63.7 |
| Spectrum 3 | 40.9 | 58.0 |
| Spectrum 4 | 23.3 | 75.7 |
| Spectrum 5 | 30.1 | 68.4 |
| Spectrum 6 | 23.0 | 76.6 |

**Figure 4.** Scanning electron microscopy (SEM) micrograph and energy-dispersive spectroscopy (EDS) for Al/Cu alloy and the results of point X-ray microanalysis of the chemical composition of the tested alloy.

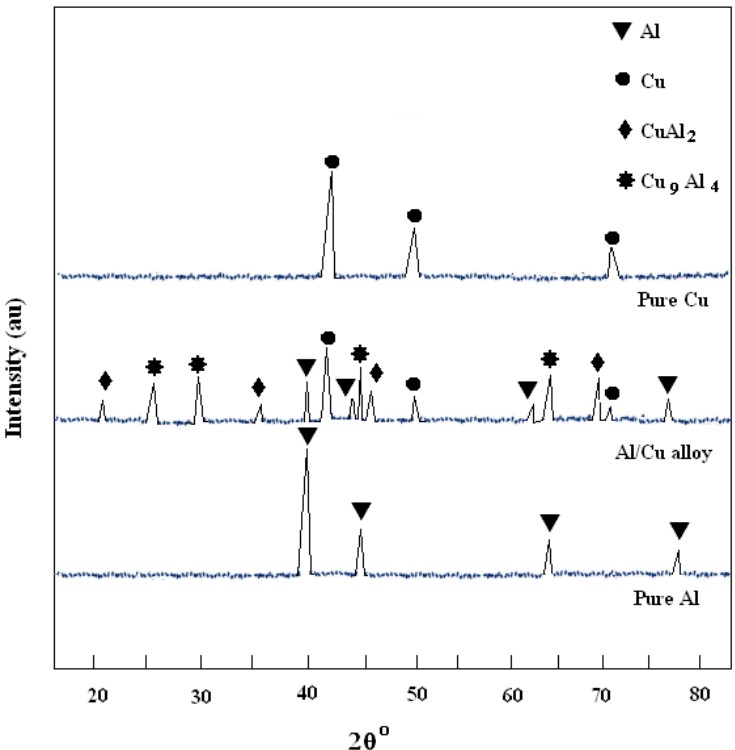

**Figure 5.** XRD diffractogram of the Al, Cu and Al/Cu alloy.

A similar mechanism of chemical reactions (4)–(6) was also proposed by the authors of the work [13]. Moreover, according to the standard Gibbs free energy of formation values for chemical reactions (4)–(6) they are all forward reactions, indicating the possibility for the formation of intermetallic compounds. The heats of formation for the phases in these chemical reactions are: $CuAl_2$: $-6.1$ kJ mol$^{-1}$, CuAl: $-5.1$ kJ mol$^{-1}$ and $Cu_9Al_4$: $-4.1$ kJ mol$^{-1}$ [27,28]. Furthermore, the heats of the formation for these intermetallic compounds can thus be arranged in order from smallest to largest as: $CuAl_2$, CuAl and $Cu_9Al_4$. Therefore, the $CuAl_2$ phase will be formed first (reaction (4)), followed by CuAl (reaction (5))

and $Cu_9Al_4$ (reaction (6)). However, in the XRD diffraction patterns for the Al/Cu alloy (Figure 5) only the $CuAl_2$ and $Cu_9Al_4$ phases were detected. Moreover, the CuAl phase was not observed. Probably, the Cu and Al atoms more readily diffuse into the $Cu_9Al_4$ phase and the CuAl phase and other are not formed on the Al/Cu surface.

### 3.2. Vickers Hardness of Materials

The hardness values of aluminum, copper and Al/Cu alloy are listed in Table 1.

**Table 1.** Vickers hardness of aluminum, copper and Al/Cu alloy surface.

| Material | Aluminum | Copper | Al/Cu Alloy |
| --- | --- | --- | --- |
| Hardness, HV20 | 29.2 ± 0.04 | 58.7 ± 0.02 | 123.3 ± 0.01 |

The Vickers hardness of the Al/Cu alloy was about 123.3 HV20, which is approximately four times greater than that of Al and about two times greater than that of Cu (Table 1). It is suggested that this strengthening of the Al/Cu alloy was affected by the presence of the intermetallic compounds such as $CuAl_2$ and $Cu_9Al_4$, which were formed through reactions ((5) and (6)) between Al and Cu during the electrical discharge alloying process. Therefore, at a higher temperature Al and Cu will react to form intermetallic compounds and these compounds have a much higher hardness, thus resulting in a significant increase in the hardness at the Al/Cu alloy.

### 3.3. Thermogravimetric Test

The thermogravimetric test was considered in registering the weight gain of the tested materials ($W = \Delta m/A$), as a function of exposure time at a certain temperature (i.e., $\Delta m/A = f(t)$) [29]. In this case the heat treatment of the Al/Cu alloy was carried out at 400 °C or 600 °C in the air atmosphere the exposure time was 9 h, Figure 6.

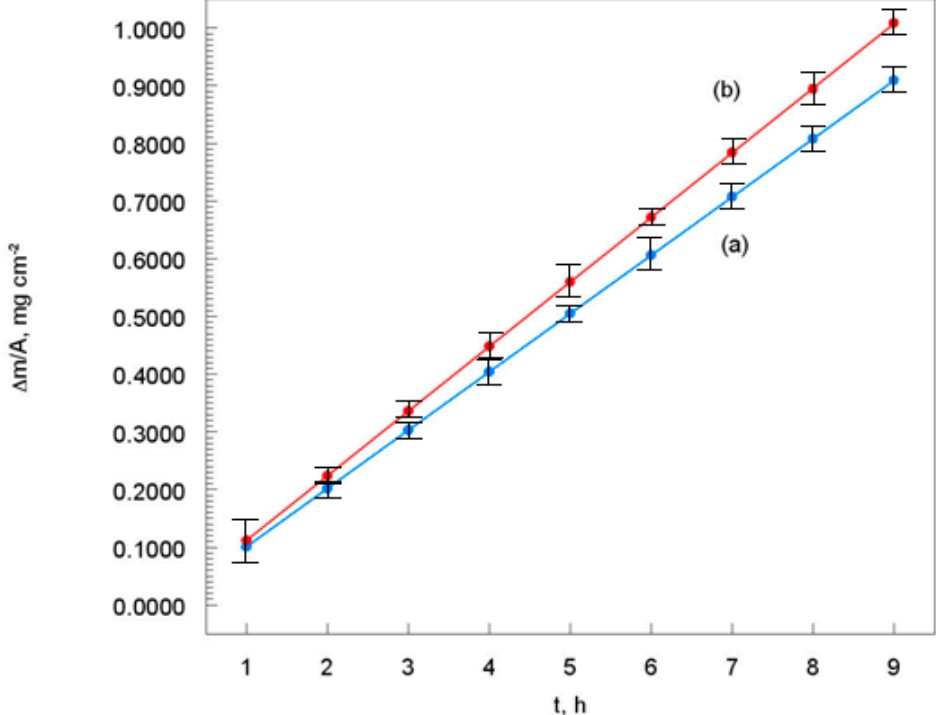

**Figure 6.** Influence of exposure time on mass change of Al/Cu alloy during heat treatment at: (**a**) 400 °C or (**b**) 600 °C in air atmosphere. Exposure time 9 h.

In temperature conditions (400 °C or 600 °C) in air atmosphere the process of chemical corrosion (oxidation) of the Al/Cu alloy proceed according to the linear law: $W = k\,t + C$, which was characteristic for the metals whose oxide surface was porous [30]. The kinetic equation that described the oxidation of the Al/Cu alloy for a temperature of 400 °C (Figure 6, curve (a)) is:

$$W = 0.10\,t + 2.08 \times 10^{-4} \tag{7}$$

However, for a temperature of 600 °C (Figure 6, curve (b)) the linear law is:

$$W = 0.11\,t + 2.78 \times 10^{-4} \tag{8}$$

Therefore, of the Al/Cu alloy surface in hot air conditions was covered with a porous oxide layer, which does not constitute an effective obstacle in the access of the oxidant to the deeper layers of the coating. The oxidation process of the Al/Cu alloy surface is described by the equations:

$$4\,Al + 3\,O_2 \rightarrow 2\,(Al_2O_3)_{ads} \tag{9}$$

$$2\,Cu + O_2 \rightarrow 2\,(CuO)_{ads}. \tag{10}$$

A complex oxide layer formed on the surface of the Al/Cu alloy, i.e., $(Al_2O_3)_{ads}$ and $(CuO)_{ads}$ (reactions (9) and (10)). Moreover, the weight gain of the Al/Cu alloy was clearly greater if the heat treatment temperature was 600 °C (Figure 6). Therefore, the oxide layer on the surface of the Al/Cu alloy that was subjected to heat treatment at 600 °C was thicker compared to the Al/Cu alloy that was subjected to heating in hot air conditions at 400 °C The effect of the heat treatment temperature on the thickness of the oxide layer on the surface of the Al/Cu alloy will be discussed later in this work.

The Vickers hardness values of the Al/Cu alloy surface after heat treatment in the air atmosphere are listed in Table 2.

**Table 2.** Vickers hardness of Al/Cu alloy surface after heat treatment.

| Heat Treatment Temperature °C | Hardness HV20 |
|---|---|
| 400 | 147.9 ± 0.03 |
| 600 | 108.8 ± 0.15 |

The Vickers hardness of the Al/Cu alloy after heat treatment at 400 °C was more than 25 HV20 higher compared to the Al/Cu alloy without heat treatment (Table 1). However, after heat treatment of the Al/Cu alloy at 600 °C the Vickers hardness of the alloy decreased significantly to 108.8 HV20 (Table 2). A drastic reduction in the hardness of the Al/Cu alloy was associated with a change in the structure of the alloy. This means that as a result of heat treatment of the Al/Cu alloy at an elevated temperature in the air atmosphere the mechanical properties of the surface of the tested material deteriorated significantly.

*3.4. Scanning Electron Microscopy Images*

Figure 7 shows the scanning electron microscopy (SEM) images of the cross-sections of the samples, before (Figure 7a) and after heat treatment at 400 °C (Figure 7b) or 600 °C (Figure 7c) in the air atmosphere. On the right side, enlarged selected fragments of Figure 7 were placed.

The cross-sections were etched with 10 wt % NaOH water solution after polishing treatment, to make the splats and microstructure in them more visible.

The Al/Cu alloy structure on the Al substrate was not homogeneous, small cracks and numerous craters were visible (Figure 7a). As a result of heat treatment at 400 °C the Al-Cu coating changed significantly. The Al/Cu alloy surface was covered with a $(Al_2O_3\text{-}CuO)_{ads}$ layer, whose thickness ranged

from 4 to 5 μm (Figure 7b). The oxide layer adhered well to the Al/Cu surface. The Al/Cu structure defects disappeared and the alloy became more compact. Therefore, the mechanical properties of the Al/Cu alloy improved. Moreover, the grains of the aluminum substrate increased significantly. After heat treatment at 600 °C the thickness of the $(Al_2O_3\text{-}CuO)_{ads}$ layer increased, reaching values from 8 to 10 μm (Figure 7c). The oxide layer cracked under the influence of elevated temperature. On the other hand of the $(Al_2O_3\text{-}CuO)_{ads}$ layer lost its protective properties with respect to the Al/Cu alloy. The thickness of the Al-Cu coating clearly decreased, but its structure did not crack. Therefore, despite the elevated temperature the Al-Cu coating did not lose its protective properties in relation to the aluminum substrate. Furthermore, the grain size increased, and the aluminum structure changed.

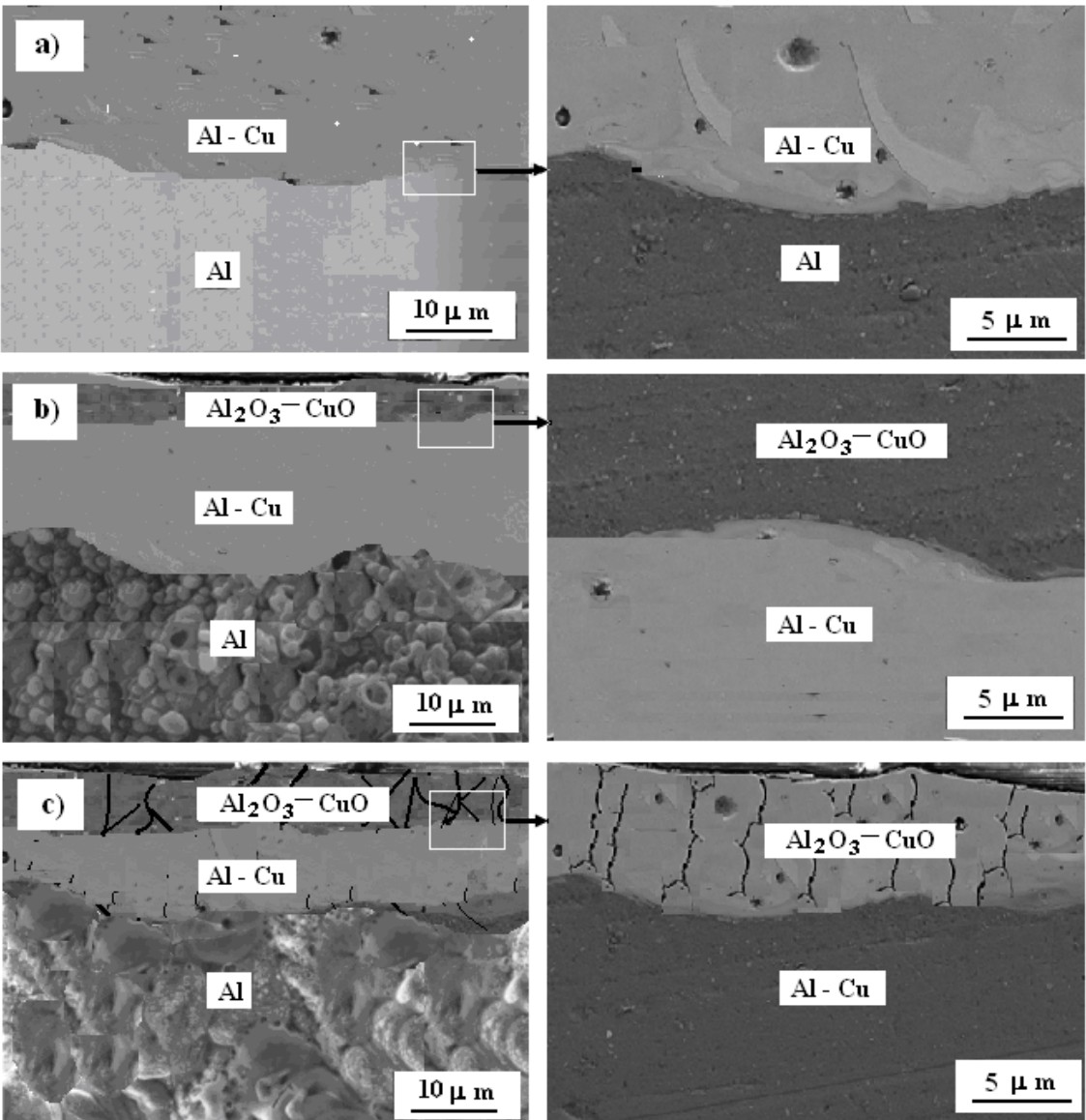

**Figure 7.** SEM images of cross-section of samples: (**a**) before and after heat treatment at: (**b**) 400 °C or (**c**) 600 °C in air atmosphere.

### 3.5. Corrosion Test

Potentiodynamic polarization (*LSV*) measurements were carried out in order to gain knowledge concerning the impact of heat treatment on the anticorrosion properties of the Al-Cu coating on the aluminum substrate and kinetics of the cathodic and anodic reactions.

The potentiodynamic polarization curves for the Al/Cu alloy were registered before and after heat treatment at 400 °C or 600 °C in the air atmosphere, Figure 8.

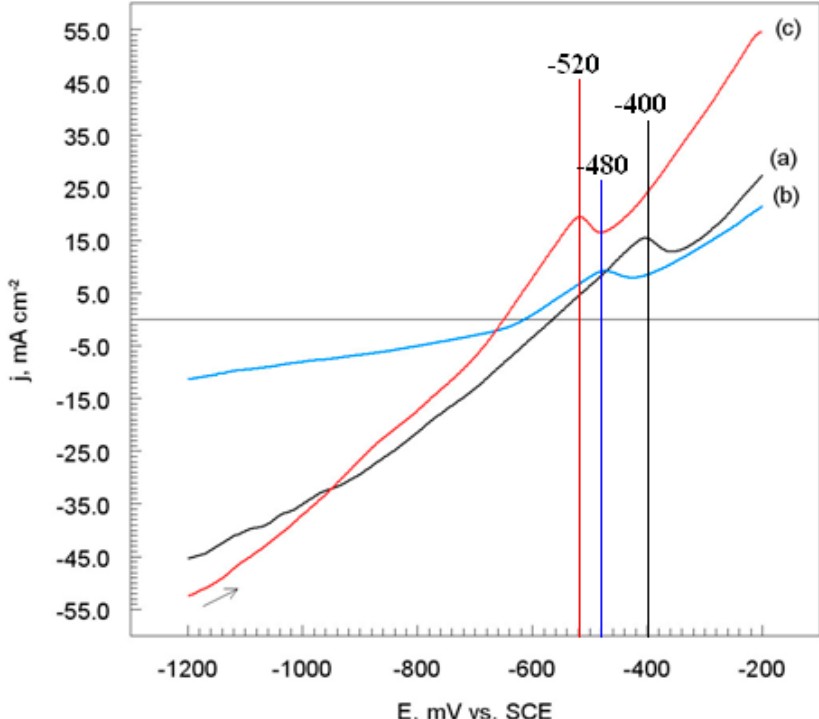

**Figure 8.** Potentiodynamic polarization curves of the Al/Cu alloy: (**a**) before and after heat treatment at: (**b**) 400 °C or (**c**) 600 °C. Solutions contained 1.2 M Cl$^-$, pH 1.5, d$E$/d$t$ 1 mV s$^{-1}$.

In the Figure 8, curve (a) refers to the Al/Cu alloy that has not been heat treated. In the acid corrosive environment the cathodic branch correspond to the reduction of hydrogen ions [2,4]. In contrast, the anode part of the potentiodynamic polarization curve relates to the active dissolution of the electrode surface. For a potential of about −400 mV vs. SCE peak appeared on the curve, which corresponds to a weak passivation process of the Al/Cu alloy surface. Thus, the electrode surface was covered with a layer of $(Al_2O_3)_{ads}$ and $(CuO)_{ads}$ oxides. In the acid chloride environment, the oxide layer dissolved, therefore the anode current density increased rapidly (Figure 8, curve (a)).

The curve (b) in the Figure 8 corresponds to the Al-Cu coating, which was subjected to heat treatment at 400 °C. However, the anode current density decreased compared to the Al/Cu alloy without heat treatment (Figure 8, curve (a)). In this case, the potential of the passivation peak shifted towards negative values, i.e., −480 mV vs. SCE. The structure of the oxide layer changed, and the $(Al_2O_3\text{-}CuO)_{ads}$ layer effectively protected the Al/Cu alloy from having contact with the aggressive corrosive environment. For a more positive electrode potential, the oxide protective layer dissolved and the anode current density increased rapidly (Figure 8, curve (b)).

The heat treatment at 600 °C of the Al/Cu alloy caused the significant disappearance of anticorrosive properties of the $(Al_2O_3\text{-}CuO)_{ads}$ coating on the alloy surface. This is evidenced by the increase in the anode current density, Figure 8, curve (c). Moreover, on the LSV curve of the passivation peak appeared for the potential of −520 mV vs. SCE. Shifting the peak towards negative potential values means degradation of the oxide layer structure. However, the oxide layer did not effectively separate of the Al/Cu alloy from the corrosive environment.

### 3.5.1. Corrosion Electrochemical Parameters

The potentiodynamic polarization curves (Figure 8) for the Al/Cu alloy produced by the electrical discharge alloying method were used to designate the corrosion electrochemical parameters before and after heat treatment in the air atmosphere. For this purpose the Tafel method was used. In this case, current density is often shown in semilogarithmic plots known as Tafel plots. Furthermore, this type of analysis is referred to as the Tafel slope analysis. In this case, we did not present a graphical form (Tafel plots) of the analysis of test results, which we put together in the form of Figure 8. Analysis results as the corrosion electrochemical parameters of the Al-Cu coating on the aluminum substrate were listed in Table 3. Along with the increase in the temperature of the heat treatment of the Al/Cu alloy, a shift (about 100 mV) of corrosion potential ($E_{corr}$) towards negative values was observed. It is noteworthy that the value of corrosive current density ($j_{corr}$) for the Al/Cu alloy after heat treatment at 400 °C clearly decreased to about 2 mA cm$^{-2}$ compared to an alloy that has not been heat treated (Table 3). As a result of thermal treatment, the corrosion resistance of the Al/Cu alloy was significantly improved. After the heat treatment of the Al-Cu coating at elevated temperature, i.e., 600 °C the corrosion current density increased three times compared to the Al-Cu coating that was treated at 400 °C (Table 3). This means that as a result of heat treatment at 600 °C corrosion resistance of the Al-Cu coating on the aluminum substrate significantly deteriorated.

The cathodic ($b_c$) and anodic ($b_a$) Tafel slope was changed as a result of an increase in the heat treatment temperature of the Al/Cu alloy on the Al substrate (Table 3). This means that the mechanism of cathodic and anodic reactions depends on the temperature of the thermal treatment of the Al/Cu alloy in the air atmosphere.

**Table 3.** Corrosion electrochemical parameters of the Al/Cu alloy on the Al substrate.

| Heat Treatment Temperature °C | $E_{corr}$ mV vs. SCE | $j_{corr}$ mA cm$^{-2}$ | $-b_c$ | $b_a$ |
|---|---|---|---|---|
| | | | mV dec$^{-1}$ | |
| Before | −564 | 2.9 | 210 | 290 |
| 400 | −614 | 2.0 | 505 | 175 |
| 600 | −650 | 6.0 | 330 | 235 |

### 3.5.2. Corrosion Rate and Polarization Resistance

The polarization resistance (Rp) of the Al-Cu coatings was determined on the basis of the slope of potentiodynamic polarization curves (Figure 8) and is summarized in Table 4.

**Table 4.** Polarization resistance and corrosion rate of the Al/Cu alloy on the Al substrate.

| Heat Treatment Temperature °C | $R_p$ (kΩ cm$^2$) | $v_{corr}$ (mm year$^{-1}$) |
|---|---|---|
| **Before** | 18 | 33 |
| 400 | 28 | 23 |
| 600 | 10 | 68 |

It was found that as a result of heat treatment of the Al/Cu surface the Rp reached values about 28 kΩ cm$^2$, which is clearly higher than the Al-Cu coating before heat treatment at 400 °C (Table 4). Thus, heat treatment at 400 °C significantly increased the anticorrosion properties of the Al-Cu coating in the aggressive chloride environment. However, an increase in temperature to 600 °C significantly reduced (three times) the anticorrosive properties of the Al-Cu coating (Table 4), which means that in this case the exchange of mass and charge between the electrode and the electrolyte solution was not slowed down during the corrosion process.

The electrochemical corrosion rate of the Al-Cu coating was calculated based on the equation (3) and listed in Table 4. For this purpose, the corrosion current density ($j_{corr}$) values of Al and Cu

were calculated depending on their content in the Al/Cu alloy (i.e., 31% Al and 68% Cu). Therefore, the corrosion rate of the Al/Cu alloy should be expressed as the sum of the corrosion rate of the alloy components i.e., aluminum and copper. The corrosion rate of the Al/Cu alloy after heat treatment at 400 °C was about one and a half lower compared to the Al/Cu alloy without heat treatment (Table 4). However, the oxide layer, i.e., $(Al_2O_3)_{ads}$ and $(CuO)_{ads}$ protected of the Al/Cu alloy surface against corrosion in the aggressive chloride environment.

On the other hand after heat treatment of the Al/Cu alloy at the temperature of 600 °C the corrosion rate increased more than three times compared to the Al/Cu alloy, which was annealed at 400 °C (Table 4). This suggests that the $(Al_2O_3\text{-}CuO)_{ads}$ layer is not tight and does not protect of the Al/Cu alloy against corrosion.

### 3.6. Chronoamperometric Measurement

Figures 9 and 10 show chronoamperometric (*ChA*) curves for the Cu coatings on the Al substrate, after heat treatment at 400 °C or 600 °C in the air atmosphere. The potentials of the working electrode were selected according to the potentiodynamic polarization curves (Figure 8, curves (b) and (c)). The exposure time of specimens in the chloride environment (1.2 M $Cl^-$) was five minutes. However, similar ChA curves were recorded for the Al/Cu alloy without heat treatment, but they were not cited in this work.

The curves (a) relate to the reduction of hydrogen ions on the surface of the working electrode [2,4]. In the case of the Al/Cu electrode after heat treatment at 400 °C an inhibitory effect of the $(Al_2O_3\text{-}CuO)_{ads}$ layer was observed. This means that the $(Al_2O_3\text{-}CuO)_{ads}$ coating was tight and the hydrogen ion reduction process was slowed down during electrolysis in the chloride environment (Figure 9, curve (a)). On the other hand after the heat treatment of the Al/Cu alloy at 600 °C a non-tight oxide layer was obtained. Therefore, the current density resulting from the hydrogen ion reduction reaction increased with increasing electrolysis time (Figure 10, curve (a)).

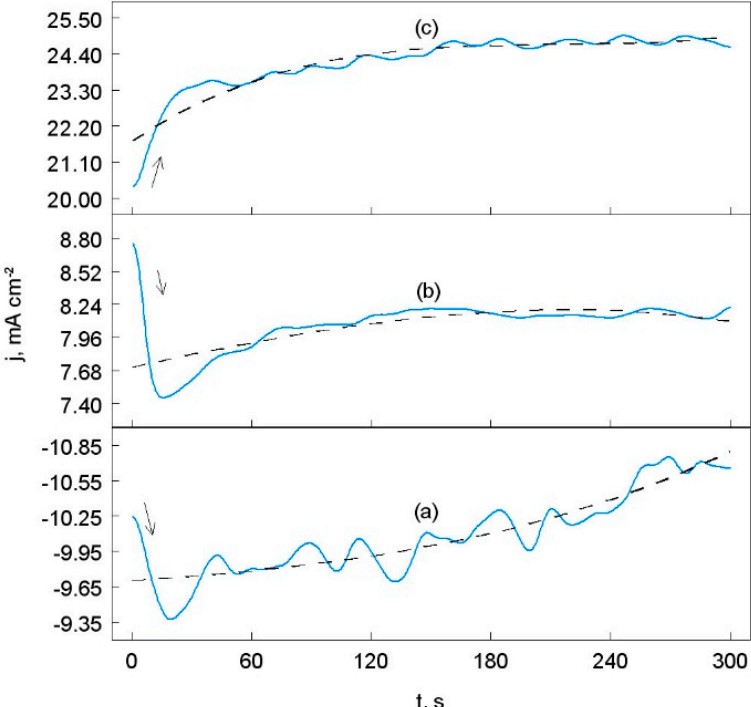

**Figure 9.** Chronoamperometric curves of the Al/Cu alloy after heat treatment at 400 °C, obtained for: (**a**) −950 mV, (**b**) −420 mV and (**c**) −300 mV vs. the saturated calomel electrode (SCE). Solution contained 1.2 M $Cl^-$, pH 1.5 (dashed lines refer to the average current density values).

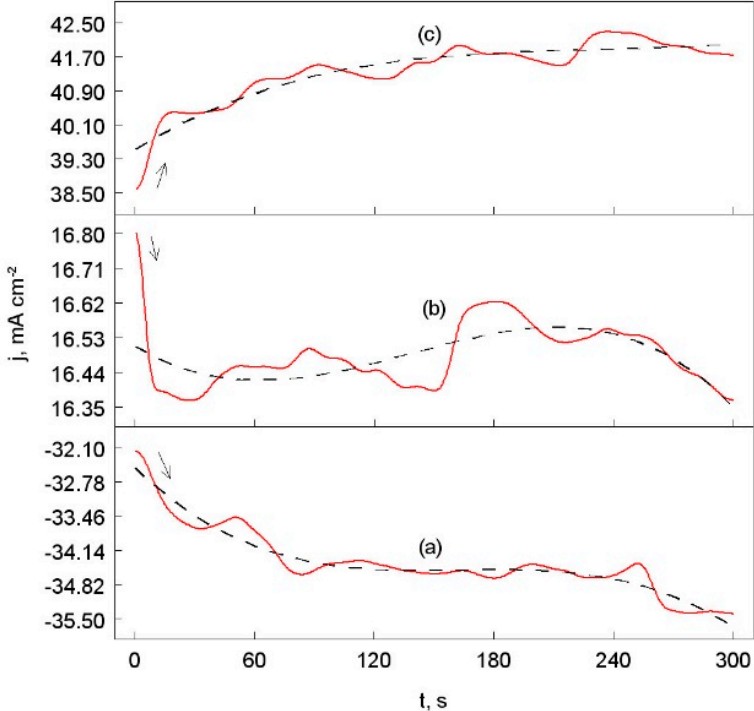

**Figure 10.** Chronoamperometric curves of the Al/Cu alloy after heat treatment at 600 °C, obtained for: (**a**) −950 mV, (**b**) −480 mV and (**c**) −300 mV vs. SCE. Solution contained 1.2 M Cl⁻, pH 1.5 (dashed lines refer to the average current density values).

The curves (b) should be attributed to the oxidation of the Al-Cu coatings. It has been observed that for the Al-Cu coating after heat treatment at 400 °C the anode current density associated with the oxidation of the electrode material remained constant (approximately 8.20 mA cm$^{-2}$; Figure 9, curve (b)). It was confirmed that the oxide layer was tight and well protected for the Al-Cu coating against electrochemical corrosion under experimental conditions. For the Al-Cu coating after heat treatment at 600 °C the anode current density that was associated with the oxidation of the alloy tested varied from 16.40 to 16.55 mA cm$^{-2}$ (Figure 10, curve (b)). So, under an elevated temperature, i.e., 600 °C, the structure of the (Al$_2$O$_3$-CuO)$_{ads}$ coating was significantly damaged and did not separate of the Al/Cu alloy before contact with a corrosive environment.

The curves (c) for the tested materials were recorded for a more positive potential value, i.e., −300 mV vs. SCE.

In this case, regardless of the temperature (i.e., 400 °C or 600 °C) of the heat treatment of the tested materials the anode current density increased systematically during electrolysis (Figures 9 and 10, curves (c)). The (Al$_2$O$_3$-CuO)$_{ads}$ was not an effective barrier to protect of the Al-Cu coating from oxidation in the aggressive chloride environment.

### 3.7. Photograph Images after the Corrosion Test

Figure 11 shows the photograph images of the Al/Cu surface produced by the electrical discharge alloying method before (Figure 11a) and after heat treatment at 400 °C (Figure 11b) or 600 °C (Figure 11c) in air atmosphere and after the corrosion test in the aggressive chloride environment. The exposure time of the specimens was five hours. To show the effects of corrosion the layer of aluminum(III) oxide and copper(II) oxide was removed from the surface of the samples. For this purpose dilute nitric acid was used and the exposure time was about three minutes. The surface of the Al/Cu alloy that was not heat treated was destroyed in an acid chloride environment as a result of electrochemical corrosion (Figure 11a).

Figure 11b shows the surface of the Al/Cu alloy (which was previously subjected to thermal treatment at 400 °C) after exposure in the corrosive environment. It is worth noting slight damage of the surface of the Al-Cu coating as a result of corrosion occurred. Therefore, the $(Al_2O_3)_{ads}$ and $(CuO)_{ads}$ layer was tight and effectively protected the metal surface from contact with the electrolyte.

After heat treatment of the Al/Cu alloy at an elevated temperature, i.e., 600 °C, the structure of $(Al_2O_3)_{ads}$ and $(CuO)_{ads}$ oxides was destroyed. Therefore, under the conditions of the experiment of the Al/Cu surface was damaged (Figure 11c).

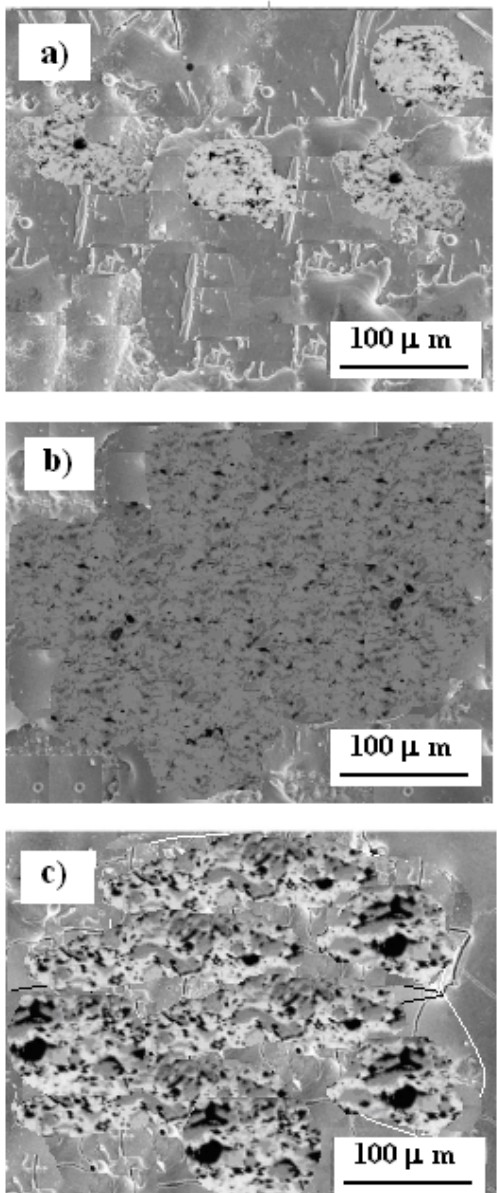

**Figure 11.** Photograph images of the Al/Cu surface: (**a**) before and after heat treatment at: (**b**) 400 °C or (**c**) 600 °C in air atmosphere and after the corrosion test in 1.2 M Cl⁻, pH 1.5. Exposure time was five hours.

The numerous pits and deep cavities were appeared on the alloy surface as a result of electrochemical corrosion.

## 4. Conclusions

1.  The aluminum-copper coating on the aluminum substrate was produced by the electrical discharge alloying method.

2.  The surface of the Al/Cu alloy contains an average of 31% Al and 68% Cu and traces of other elements.

3.  The mechanical properties of the Al-Cu coating after heat treatment in the air atmosphere significantly increased or decreased at 400 °C or 600 °C, respectively.

4.  After heat treatment at 400 °C the oxide layer, i.e., $(Al_2O_3)_{ads}$ and $(CuO)_{ads}$ protected of the Al/Cu alloy surface against corrosion in the aggressive chloride environment.

5.  However, after heat treatment at an elevated temperature i.e., 600 °C, the oxide coating structure was destroyed. Therefore, the mechanical properties of the Al/Cu alloy decreased and its surface had undergone deep electrochemical corrosion.

**Author Contributions:** Conceptualization, M.S. and S.S.; formal analysis, M.S. and K.S.-S.; investigation, K.S.-S. and P.M.; methodology, M.S.; writing—original draft, M.S. and K.S.-S.; writing—review and editing, M.S. All authors have read and agreed to the published version of the manuscript.

**Funding:** These studies were not financed from outside.

**Acknowledgments:** The authors thank Dominica Kaminska for cooperation and participation in experiments.

**Conflicts of Interest:** The authors declare no conflict of interest.

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
