# Peer review of "Influence of Heat Treatment on the Corrosion Resistance of Aluminum-Copper Coating"

_metals, doi:10.3390/met10070966_

Round 1

Reviewer 1 Report

The coating technology used of this paper is interesting. The authors made corrosion tests in a very acidic media and they proposed a ranking from the corrosion results. With all the corrosion rates of 10 mm/year it is difficult to propose a classification. Also the document contains some mis interpretation of the electrochemical polarisation diagrams (see the comments in the enclosed document).

Author Response

Reviewer 1

Thank you very much for the thorough and professional evaluation of our article. Below I have written the responses to the comments and suggestions that are contained in the review.

  • (21) after the corrosion test ?
  • (27) Not clear : What is green?
  • (77/78) Sentence not complete
  • (106) not clear
  • (167/168) What Tafel values have been used in this formula?
  • (256) Where have been performed the HV measures: on a cross section or on the surface?
  • (297/299) Wrong interpretation: the current peak corresponds to the active dissolution. After more positive values of the potential you can have the passivation of the alloy but in your results the current density is very high (>10 mA/cm2) and corresponds to an anodic dissultion due to the acidity of your electrolyte.
  • (302/318) As indicated in the previous comment your interpretation of the peak values is erroneous.
  •  
  • (344/345) Table 4) Even for the 400°C HT sample, the corrosion rate is very high and not at all in agreement with a good corrosion protection.
  • (370/372) You should indicate versus what these potentials values are given: mV/SCE.

Response: I corrected the text of our article in accordance with the comments and suggestions of the reviewer. Moreover, I have rewritten the discussion of the measurement results, which I included in the article in the form Figure 8. i.e. Potentiodynamic polarization curves of Al/Cu alloy: a) before and after heat treatment at: b) 400 °C or c) 600 °C. Solutions contained 1.2 M Cl-, pH 1.5, dE/dt 1 mV s-1.

Again, I thank you very much for all your efforts in improving our manuscript. We hope now that our manuscript is suitable for publication.

Reviewer 2 Report

This paper studies the influence of heat treatment on the corrosion resistance of the aluminum/copper (Al/Cu) coating on the aluminum substrate The Al/Cu alloy was produced by electrical discharge alloying (EDA) method. the microstructure, mechanical properties and corrosion resistance of the materials were carried out Before and after heat treatment. The present paper is interesting, however, to be accepted for publication the following comments need to be addressed

  • The introduction section needs to be improved by citing new and related articles of the current journal.
  • The proficiency of the language needs a more improvement in the manuscript
  • The conclusion part should be more concise

The methods adequately described, and the results clearly presented. the manuscript can be accepted after minor revision

Author Response

Reviewer 2

Thank you very much for the thorough and professional evaluation of our article. Below I have written the responses to the comments and suggestions that are contained in the review.

  • The introduction section needs to be improved by citing new and related articles of the current journal.

Response: As suggested by the reviewer in Chapter 1. Introduction, I cited four new articles [14, 15, 16, and 17], the content of which concerns the subject of our article.

  • The proficiency of the language needs a more improvement.

Response: Our manuscript will be subject to intensive improvement in terms of English. To this end, we will use a professional English editing service.

  • The conclusion part should be more concise.

Response: As suggested by the reviewer, we have shortened Chapter 4. Conclusions.

Again, I thank you very much for all your efforts in improving our manuscript. We hope now that our manuscript is suitable for publication.

Reviewer 3 Report

The paper is an interesting study about the influence of the heat treatment on the corrosion resistance of a coating based on an aluminum/copper alloy.

-I think that English must be revised, because the manuscript is difficult to be read.

-The authors report the influence of the thermal treatments on the mechanical properties. I think that in addition to the alloy hardness, a study on the tensile, compressive and flexural strenghts should be reported in order to understand the effect of these treatments. Which effect on the ductility of the materials has been obtained?

-SEM images of figure 7 must report a magnification of the Al/Cu//Al and Al2O3/CuO//Al/Cu interfaces in order to better show the adherence of the layers. Moreover, in fig 7C a magnification of the cracks should be reported also to better explain why they show similar thickness.

For this reason, major revisions are required.

Author Response

Reviewer 3

Thank you very much for the thorough and professional evaluation of our article. Below I have written the responses to the comments and suggestions that are contained in the review.

  • I think that English must be revised, because the manuscript is difficult to be read.

Response: Our manuscript will be subject to intensive improvement in terms of English. To this end, we will use a professional English editing service.

  • The authors report the influence of the thermal treatments on the mechanical properties. I think that in addition to the alloy hardness, a study on the tensile, compressive and flexural strengths should be reported in order to understand the effect of these treatments. Which effect on the ductility of the materials has been obtained?

Response: The reviewer is right. In addition to hardness, tests to determine the tensile, compressive and flexural strengths of the tested materials should be carried out to determine the mechanical properties. Unfortunately, in my laboratory we do not have adequate equipment to examine all parameters regarding the mechanical properties of materials.

  • SEM images of figure 7 must report a magnification of the Al/Cu/Al and Al2O3/CuO//Al/Cu interfaces in order to better show the adherence of the layers. Moreover, in figure 7C a magnification of the cracks should be reported also to better explain why show similar thickness.

Response: Thank you to the reviewer for your valuable attention regarding Figure 7. Selected fragments of SEM images of cross-section of samples (Fig. 7) were enlarged to better show the change of coating properties as a result of thermal treatment of materials.

Again, I thank you very much for all your efforts in improving our manuscript. We hope now that our manuscript is suitable for publication.

Reviewer 4 Report

The manuscript by Scendo et al. discusses about the processes of obtaintion of coatings with corrosion resistance by electrical discharge allowing of Cu over Al. This is a topic that is fairly well studied, but still has considerable interest. The article present some interesting data, but I mean that it at the present state is too shallow to allow publication in Metals. I offer some comments below that the authors may want to consider.

  • In the abstract Author should avoid conclusions as”the surface did not corrode”
  • In Fig 1 there are not described parameters.
  • Author should avoid expressions like “thin coating, because this is a terminology for coating of less than 1 micron.
  • The results presented in Fig 4 may be improbed with measurements in the clearest zone
  • Author have to justify why only appears Cual2 and Cu9Al4, and no other phases.
  • Please revise the hardness units
  • The words and numbers of the figures are too small

Author Response

Reviewer 4

Thank you very much for the thorough and professional evaluation of our article. Below I have written the responses to the comments and suggestions that are contained in the review.

  • In the abstract Author should avoid conclusions as “the surface did not corrode”

Response: I deleted the unfortunate fragment of the sentence and replaced it with a new one “the oxide layer well protected of the ally surface against corrosion”

  • In Fig. 1. there are not described parameters.

Response: I supplemented the description of the symbols in Figure 1, which shows a scheme for the production of coatings using the EDA method.

  • Author should avoid expressions like “thin coating” because this is a terminology for coating of less than 1 micron.

Response: Thank you very much to the reviewer for your valuable attention. The unfortunate term thin coating was deleted in the article.

  • The results presented in Fig. 4. may be improved with measurements in the clearest zone.

Response: Figure 4 concerns scanning electron microscopy (SEM) micrograph and energy-dispersive spectroscopy (EDS) for Al/Cu coating and the results of point X-ray microanalysis of the chemical composition of tested alloy, which I have listed in the table next to the figure. Moreover, I selected the most characteristic places on the surface of the cross-section of the sample for measuring the chemical composition of the alloy. However, I did not specify the chemical composition of the substrate in the table, because I received a result of 100% Al.

  • Author have to justify why only appears CuAl2 and Cu9Al4, and no other phases.

Response: In the XRD diffraction patters for the Al/Cu alloy (Figure 5) only the CuAl2 and Cu9Al4 phases were detected. The CuAl and other phases were not detected. I suppose there are two reasons that caused us to detect only two phases:

  • The Cu and Al atoms more readily diffuse into the Cu9Al4 phase and the CuAl phase and other are not formed on the Al/Cu surface.
  • For our research, we have already used an outdated X-ray spectrometer (Carl Zeiss Jena, type HZG4). It seems to me that the X-ray spectrometer has a low detector sensitivity and therefore we have not detected the other phases (except CuAl2 and Cu9Al4) on the surface of the Al/Cu alloy.

  • Please revise the hardness units.

Response: I have improved the unit of hardness of materials on the Vickers scale according to the scheme: xxHVyy, where x is the hardness number, HV gives the hardness scale (Vickers), and y - indicates the load used in kgf.

  • The words and numbers of the figures are too small.

Response: I have increased the dimensions of all figures so that they can be easily interpreted.

Again, I thank you very much for all your efforts in improving our manuscript. We hope now that our manuscript is suitable for publication.

Round 2

Reviewer 1 Report

The authors always refer to passivation peak. In this media the peak is an active dissolution peak with for more positive anodic potential a decrease to always very high anodic current densities (more than 5 mA/cm2) : it is a very limited slow down of the dissolution of the coating but not a real passivation.

Reviewer 3 Report

The paper can be now accepted but the authors must report that only hardness measurements instead of mechanical tests were carried out.